# Data Acquisition Filtering Focused on Optimizing Transmission in a LoRaWAN Network Applied to the WSN Forest Monitoring System

**DOI:** 10.3390/s23031282

**Published:** 2023-01-22

**Authors:** Thadeu Brito, Beatriz Flamia Azevedo, João Mendes, Matheus Zorawski, Florbela P. Fernandes, Ana I. Pereira, José Rufino, José Lima, Paulo Costa

**Affiliations:** 1Research Centre in Digitalization and Intelligent Robotics CeDRI, Instituto Politécnico de Bragança, 5300-252 Bragança, Portugal; 2Laboratório para a Sustentabilidade e Tecnologia em Regiões de Montanha (SusTEC), Instituto Politécnico de Bragança, 5300-252 Bragança, Portugal; 3INESC TEC—INESC Technology and Science, 4200-465 Porto, Portugal; 4Faculty of Engineering, University of Porto, 4200-465 Porto, Portugal; 5Algoritmi Research Centre/LASI, Campus Azurém, University of Minho, 4800-058 Guimarães, Portugal

**Keywords:** data transmission optimization, wireless sensor network, wildfire, LoRaWAN, Internet of Things, digital filter

## Abstract

Developing innovative systems and operations to monitor forests and send alerts in dangerous situations, such as fires, has become, over the years, a necessary task to protect forests. In this work, a Wireless Sensor Network (WSN) is employed for forest data acquisition to identify abrupt anomalies when a fire ignition starts. Even though a low-power LoRaWAN network is used, each module still needs to save power as much as possible to avoid periodic maintenance since a current consumption peak happens while sending messages. Moreover, considering the LoRaWAN characteristics, each module should use the bandwidth only when essential. Therefore, four algorithms were tested and calibrated along real and monitored events of a wildfire. The first algorithm is based on the Exponential Smoothing method, Moving Averages techniques are used to define the other two algorithms, and the fourth uses the Least Mean Square. When properly combined, the algorithms can perform a pre-filtering data acquisition before each module uses the LoRaWAN network and, consequently, save energy if there is no necessity to send data. After the validations, using Wildfire Simulation Events (WSE), the developed filter achieves an accuracy rate of 0.73 with 0.5 possible false alerts. These rates do not represent a final warning to firefighters, and a possible improvement can be achieved through cloud-based server algorithms. By comparing the current consumption before and after the proposed implementation, the modules can save almost 53% of their batteries when is no demand to send data. At the same time, the modules can maintain the server informed with a minimum interval of 15 min and recognize abrupt changes in 60 s when fire ignition appears.

## 1. Introduction

Wildfires destroy millions of hectares of vegetation yearly, causing numerous losses of human lives, fauna and flora, and environmental contamination. The origin of forest fires may have different sources, natural or not. Typically, only around 4% of all forest fires have natural causes, such as meteorological factors. In all other cases, humans are responsible for the fires, deliberately or due to negligence [1]. Moreover, forests are generally remote, abandoned/unmanaged areas composed of high-density biomass (for example, rotten trees, leaves, and bushes, among others). As if that was not enough, everything in a forest can be transformed into fuel for fire ignition during dry seasons [2].

Forest fires are becoming a natural part of the European landscape as they are increasing in size, occurrence, and severity, mainly due to the combination of climate change and human activity [3,4]. In 2022, the annual technical report by the Joint Research Centre (JRC), the European Commission’s science and knowledge service, indicates that from June to mid-September, the worst forest fires in Europe once again occurred in countries bordering the Mediterranean Sea [5]. The fires were attributed to high summer temperatures, strong winds, a prolonged dry season, and deforestation. As a result, over the years, these fires have been widespread and had a devastating social and economic impact. Uncontrolled wildfires are an immediate danger to people and animals and cause air pollution, which can exacerbate respiratory illnesses. Wildfires also release large amounts of carbon dioxide into the atmosphere, contributing to climate change.

Although the Mediterranean Sea does not bathe Portugal, due to its geographical proximity, it exhibits climatic characteristics similar to other Mediterranean countries. During the summer, Portugal also tends to register a dry climate for prolonged periods and strong winds. This similarity has made Portugal subject to numerous forest fires in recent years [6]. To solve the problem, the government implemented measures such as aerial firefighting, increased surveillance, and forest management [7]. However, some of the plans developed to combat forest fires may not have a high success rate in mountainous regions, such as in the district of Bragança, an inland region in the northeast of Portugal. For example, surveillance systems using cameras (such as [8]) can suffer from block view when monitoring forests in mountainous areas. In this sense, if a forest ignition starts in a valley between two mountain ridges, depending on the camera position, this forest ignition will not be seen in a short time interval.

Regarding the risk and uncertainty of this type of environment, developing strategies to monitor forests focusing on avoiding catastrophe is an urgent and crucial task, especially in Portugal, one of the European countries that most suffer from fire forest damages [1]. An efficient monitoring system can be precious in reducing the forest fire risk, supporting decision-making, and reducing time response when ignitions are detected. The time response is essential due to the spreading of fire potential. Therefore, early detection is critical to the safety and security of environmental zones. It is, in fact, one of the most vital challenges for the government and forest fire managers [9]. According to [10,11] and also to Portuguese fire authorities, the maximum time interval from ignition to firefighters’ alert response should be at most 15 min. Otherwise, the fire will be out of control due to the fire’s fast propagation speed.

The possibilities and challenges in developing a robust surveillance system are many. In this sense, the new features inserted in Forest 4.0 can digitize regions to help combat teams in decision-making. The Forest Alert Monitoring System (SAFe) project aims to build a Wireless Sensor Network (WSN) in the Serra da Nogueira forest, located in the northeast of Portugal [12]. The system proposed is divided into three main stages, as illustrated in Figure 1: 1—data acquisition in the forest, 2—data processing on the cloud, and 3—alert generation for the authorities. First, the sensor modules are strategically allocated in the forest for data acquisition (demonstrated in [13]) and pre-processed by the algorithms presented in this work.

A network composed of low-cost module sensors is designated for data acquisition and sends the data to a cloud system to be processed for fire detection in minimal time by autonomous evaluation. The system communication is based on LoRaWAN technology. Even though LoRaWAN targets low energy consumption scenarios, the modules still need to save energy during data transmission. In this work, considering the interval established in the previous work [14], the current measurement of the consumption of each module is identified, and the total time that a battery can support. This is because they will be placed in remote locations, and the longer their batteries last, the more robust the WSN will be in terms of autonomy. In this sense, it is essential to develop a strategy to optimize the communication between the sensor modules and the cloud since the moment of transmission requires a current peak during the battery consumption.

In this approach, the data will be strictly analyzed in the cloud to identify outliers, false alerts, cross information between the modules, and confront the current data collected with historical data storage. Thus, when any module detects a fire ignition, an alert notification should be immediately sent to the authorities by a user interface. The user interface includes an online map that shows the position of the sensors and devices, as well as a dashboard with graphs and other visualizations to monitor the data. In addition, the user can access a library of fire prevention and mitigation resources. Thereby, four algorithms were developed to optimize the communication between the sensor modules and the cloud system: an exponential smoothing algorithm (A1), two strategies designated as the moving average-based algorithms (A2 and A3) and a least squares based algorithm (A4). These algorithms were designed to require low memory capacities and low computing power since they will be embedded in the ATmega328 microcontroller, used as the main PCB’s core [15].

This paper is organized as follows: after the introduction, Section 2 presents the LoRaWAN communication system and some studies and applications of this technology in forest fire monitoring; the SAFe project system is presented in Section 3, and the data acquisition process and communication problems are treated in the same section. The proposed strategies and algorithms to optimize the LoRaWAN communication are presented in Section 4; thereafter, the results obtained by the use of these algorithms in day and night conditions are shown in Section 5; finally, Section 6 concludes the paper and points out some future work directions.

## 2. Related Work

Once forests usually are large areas of difficult access, and spreading sensors along forests pose several challenges, namely regarding power supply and data transmission. Due to the areas’ dimensions that need to be covered, Low Power Wide Area Network (LPWAN) technologies [16,17], such as Long Range (LoRa) [18,19], SigFox [20,21], Long Term Evolution (LTE), or 5G [22,23,24,25,26], are reasonable solutions [27,28]. However, LoRa technology is earning popularity since it provides efficient communication with good battery lifetime, capacity, range, and cost [28,29].

LoRa is a physical layer that enables a long-range communication link, and LoRaWAN is a communication protocol and system architecture for the network designed to support a large part of the billions of devices associated with the Internet of Things (IoT) [29,30,31]. The Things Network (TTN) [32], which supports LoRa modulation focused on LoRaWAN, is a service that provides a set of open-source tools and a global network to build IoT applications at low-cost, featuring maximum security and being ready-to-scale [33]. The availability of this type of transmission allows the development of a real-time monitoring system in previously unthinkable places, such as forests.

The use of LoRa in forest monitoring systems is explored in [28]. In this case, a system comprises several LoRa nodes with sensors to measure the temperature, relative humidity, wind speed, and CO${}_{2}$ of the environment. The data are sent by LoRa using the TTN server, being processed and analyzed by a cloud-based server. A similar approach is presented in [34], which describes a low-power wildfire detection system with the capability of remote monitoring and data transmission by LoRa technology. The proposed system consists of a sensing head with a wireless node, a gateway node, and the central controller where the information and the decision algorithms process the data to identify the presence or absence of fire [34]. In both works, [28,34], the modules’ power supply is performed by batteries connected to a solar panel. Moreover, in these systems, it is possible to apply efficient programming to use sleep routines, and consequently, no significant variations in the sensors data are detected [28].

A low-cost smart system of wireless sensors based on narrow beam Far Infrared (FIR) is presented in [33]. This system can detect strong heat sources in real-time, such as fire. The authors propose using the developed system in combination with active fire protection mechanisms, such as water sprinklers, to attempt to sustain the advance of the flame front. The modules’ communications with the gateway are once again in charge of the LoRa method, sending constant messages with 15 min time intervals and alerts whenever the sensors have abnormal reading values.

With CO and NO gas sensors, the MiCS-4514, the work in [35] proposes the use of a wireless mesh network to identify fires using the LoRaWAN network. In addition to the gas sensor, the modules have an LM35 temperature sensor. The modules communicate through a Gateway based on Raspberry Pi, which will send the data to an interface connected to a MySQL database. Another project, presented in  [36], applies Arduino Mega2560 as its main core. This platform supports an MQ-135 sensor (NH3, NOx, alcohol, Benzene, Smoke, and CO2.), DHT-22 (humidity and temperature), and a Dragino Lora Shield (LoRa transceiver). Then, with this preliminary node, which does not have photovoltaic panels, the authors propose to create a WSN to identify fire ignitions. The results of the tests carried out in [37] indicates that the LoRaWAN technology is an excellent candidate, in terms of response time and network delay, for the installation of fire detection and prevention systems in smart buildings. The mentioned work uses 32 nodes, forming a WSN, which detects smoke, gases (liquefied petroleum gas, propane, and methane, among others), temperature, and humidity. The authors do not use real situations of fire.

Most of the approaches that apply LoRa in forest monitoring systems use a solar panel to charge the battery or algorithm resources to select the data transmission and save energy. As mentioned before, the SAFe project’s main objective is to develop low-cost sensors and spread them in large quantities in the forest. Thus, adding a solar panel will increase the final cost of implementing the SAFe project. Furthermore, in several situations, the modules will be fixed at trunk trees and, consequently, the treetop will cover (full or partial) the sunlight, in which case, the solar panel could not work as expected. In addition, not only can trees shade the modules’ solar panels, but mountains can also drastically slow down battery charging through the solar panel. This is because, in certain circumstances, such as modules fixed between two mountain peaks, they will receive little exposure to sunlight due to their relief characteristics.

Considering these three cases, such an alternative to installing solar panels in each module was discarded. Thus, the non-application of solar panels in the modules will result in periodic maintenance. Periodic maintenance can be accepted in cases of long duration since short durations result in higher costs for the government to hire people and/or equip employees to access difficult-to-access areas (a typical characteristic of mountainous regions or dense forests). Therefore, it was necessary to implement an alternative to reduce the batteries consumption: some strategic algorithms were developed to optimize the data transmission and, consequently, increase the lifetime of the battery’s modules; these algorithms are the main contribution of this paper, and they will be presented in the next sections.

## 3. SAFe Project System and Data Transmission

As mentioned in the previous section, the SAFe project aims to build a WSN in regions with fire ignition potential. The system architecture includes a range of innovative tools and activities, which are designed to reduce the risk of fire ignition and to improve the effectiveness of fire prevention activities. The architecture, illustrated in Figure 2, is designed to integrate all the components in a single system, allowing the information to be exchanged and accessed easily. Thus, the SAFe project also provides an interactive user interface that allows users to access and interact with the system efficiently and intuitively.

The SAFe architecture consists of four main components: a monitored region, a set of sensor modules, a communication system, and a control center. Together with a management system based on artificial intelligence, these four elements will enable efficient and intelligent data analysis, generating forest ignition alerts, warning rescue, and combat teams such as firefighters, civil protection, or city hall. These elements can be further broken down into eight categories that work together. These eight categories are illustrated in Figure 2 and are next described:The monitored region (**➀**) is where the WSN will be placed to collect data. The choice of these regions should consider the annual fire risk map provided by the Instituto da Conservação da Natureza e das Florestas (ICNF) [38];The WSN (**➁**) is responsible for real-time data acquisition at the forest. The Sensor Module allocation is determined through an optimization procedure that evaluates the fire hazard in each coordinate and must also consider the forest characteristics, such as soil type, cover tree density, and terrain relief, among others [13,39];The LoRaWAN Gateway (**➃**) receives data from each sensor module via the LoRaWAN protocol (**➂**). Then, it forwards the data through a 4G/LTE link (**➄**)—or by Ethernet where available—to a cloud server (**➅**);The Control Center (**➆**) receives all information, computes the data, and sends alerts about hazardous situations or forest fire ignitions, to the surveillance agent in the region. Therefore, this control center has an associated server (**➅**) that stores all collected data over the years and performs artificial intelligence procedures which will generate forest ignition alerts, warning rescue and combat teams;When rescue and combat teams receive the alerts provided by the algorithms, it becomes possible to elaborate an attack strategy as a support decision. Specifically, they can act against wildfires knowing the precise positioning of fire ignitions (**➇**).

This work focuses only on data transmission using LoRaWAN protocol, explicitly pre-filtering data read from sensors before sending it to the server. The approaches pursued by SAFe regarding the other architecture components are described in [12,13,14,15,39]. As such, considering the high number of nodes used in the SAFe project to monitor the target region, this work will identify a generic digital filter that can be used in the firmware of all modules. Therefore, it is expected that the filtering process conducted before data transmission will save energy for each module (due to the current peak during transmission) and save bandwidth on the LoRaWAN network (by removing unnecessary messages). The following subsection will demonstrate the energy consumption of a model of the sensor module used in the SAFe WSN.

### Consumption’s Identification

In previous work [14] the full data package is described (battery level, flames sensors, temperature, and humidity) and, it is demonstrated that the data package with all sensor values of each module can be sent using more than 60-second time intervals (in this case, the minimal time interval is 60 s). Using this configuration, each module can send data for almost 45 days, the Figure 3 shows the discharge battery from the high level to the low level in laboratory tests.

Based on this battery discharge, it is necessary to identify the energy consumption during data transmission through the LoRaWAN protocol. Thus, the device Power Profiler Kit II (nRF-PPK2) from the Nordic company is used [40]. This device can measure current in the range of 200 nA to 1 A under a continuous voltage ranging from 0.8 V to 5 V. Moreover, the nRF-PPK2 supports various features for measuring low sleep current, high active current, and short current peaks. It includes the Power Profiler application, which allows users to monitor and measure the device’s power consumption in real time. The Power Profiler application can be used to measure both the active and sleep currents of a given PCB. It can also detect current peaks, allowing users to know how much current is used during certain operations.

It is possible to define the PCB consumption used in SAFe’s WSN nodes with the nRF-PPK2 as the power supply, which means that the battery will be removed in a swap for VOUT and GND pins of the nRF-PPK2. Then, the power profiler application is set up to supply a voltage of 3.7 V (the same as the model 18,650 battery [15]), 60 s to perform the measurement (the interval between each data transmission), and a data sampling of 100,000 per second. Note that the LoRaWAN used in each module is based on Classe A, which has the lowest energy consumption among other class types. Moreover, despite the chance each module can receive control signals, the measurement presented here will not consider consumption while receiving messages.

After the specified interval, it is possible to analyze the PCB consumption in each stage of its operation loop. Figure 4 demonstrates this entire measurement range performed with the nRF-PPK2. It is possible to visualize three main operations, namely: the deep sleep intervals of the node, the moment of data acquisition, and the instant of data transmission. As the moment when nodes are in a deep sleep is not the focus of this work, only the other two moments were selected in gray. Thus, two signal analyses are in the graph generated by the power profiler application. The first one demonstrates the PCB consumption during the 60 s, with an average consumption of 672.30 μA. In addition, the second analysis indicates the consumption only in part highlighted in gray (sensor reading and data sending), with an average of 10.57 mA.

The peak current measured during the interval highlighted in gray shows how much consumption each module uses to send data to the server. Consequently, sending data every 60 s is not a satisfactory alternative if no significant variations exist. For example, in the case that the flame, temperature, and humidity sensor values of a given sensor module are unchanged for 300 s, this module would consume its own battery and LoRaWAN network bandwidth to inform unnecessary data to the server. This fact would get worse over the days and weeks, causing the battery life to be drastically reduced. In addition, using more bandwidth of a LoRaWAN Gateway implies increasing the number of gateways and/or decreasing the number of SAFe WSN modules.

It is not enough to check between each operation loop that the sensor values are identical to those read in a previous one. This is because the environment where the sensors are fixed can change naturally. In a basic case, when a breeze passes, slight temperature variations can be confused as fire ignitions using the direct difference between actual and previous data. Therefore, developing a digital filter that will select only the critical data to be sent to the server is necessary. In addition, it is also essential to specify the behavior of anomalies in the data acquisition, which may indicate the appearance of forest ignitions.

Since algorithms pre-process these last stored data, this filter can still be based on the difference between actual and previous data sensors inserted in each node. In this sense, the algorithms can eliminate/decrease the amount of noise that the environment generates. On the other hand, this filter must respect the computational limits and free memory available of the ATmega328 microcontroller. Therefore, the algorithms inserted in the filter cannot have high computational complexity. Moreover, it will not be possible to store long-term data (the volume of data at each read loop is shown in [15]). Furthermore, the filter needs to benefit all the modules developed; that is, it needs to be generic. Otherwise, configuring each module with different variables will be arduous when the WSN becomes dense. In this regard, the next section will describe each approach used to develop the Data Re-send Only Warnings Sensing an Ignition Else Repose (DROWSIER) filter.

## 4. Algorithms for Data Transmission Optimization

Since the microcontroller used in each module (ATmega328) has low memory available after the firmware’s implementation, the DROWSIER filter needs to have a modest memory footprint and yet still be effective and computationally efficient (these constraints do not apply to the algorithm decision stage, once most of its operations are executed in the cloud-based server). In this sense, the algorithms presented in this section focus on low computational processing through basic math operators and a small number of variables.

Four approaches were developed to solve the peak consumption problem during communication, as pointed out before, through low-complexity algorithms. One approach, named A1, uses exponential smoothing to identify peaks during the data acquisition. Two approaches, A2 and A3, use the moving average techniques during a real-time comparison between historical values stored in arrays. Similarly, the last approach, A4, is developed using the prediction of through the behavior of least squares. Each of them is described in the following subsections.

### 4.1. Exponential Smoothing Algorithm

Exponential smoothing is used in various fields, such as economics [41], engineering [42,43,44], and weather forecasting [45]. It is a way to reduce the noise in a time series data set and make it easier to identify trends and long-term patterns in the data. The basic idea behind exponential smoothing is to give more weight to recent observations than to observations from a defined period of the past [46,47].

The weighting factor is usually set to an exponentially decreasing value. Using this decreasing exponential weighting, the most recent data points in the time series will be given the highest weight, and the oldest data points will be given the lowest. It helps to reduce the effect of outliers or spikes in the data and to provide a more consistent view of the trend in the data. This ensures that the most recent trends and patterns are captured while also allowing for some noise and random variation [46,47]. As such, exponential smoothing can be used to represent the underlying trend and forecast future values more accurately. Several variations on the basic exponential smoothing technique exist, including single, double, and triple exponential smoothing [48].

The technique can smooth out the noise in time series data, working as a low-pass filter to remove high-frequency noise. Therefore, this technique is applied to identify sudden changes caused by a forest ignition. Thus, this approach may improve the communication of each module by identifying short anomalies during data acquisition. The approach using exponential smoothing will be named A1 and is based on Equation (Equation 1): (1)$$Y_{t}=\omega X_{t}+\left(1-\omega \right)Y_{t-1},\;t>0$$

Here $Y_{t}$ is the output after applying an exponential smoothing filter, *t* is the time, ω is a weight variable in 0<ω<1, $X_{t}$ represents the actual data given a *t*, and $Y_{t-1}$ is previous data given a *t*.

This A1’s equation can return outputs in several time intervals; that is, it is possible to specify the window function in the time series data set to remove peaks in its values. These peaks are exhibited in Figure 5, where there is a graph with the values stored from the five flame sensors of a module in the interval from 4:00 am to 10:00 pm on a day with a fire test.

Over the indicated period, it can be observed that there are several peaks during the data reading. Nonetheless, there are different behaviors of these peaks. For example, in the samples from 1:00 pm to 2:00 pm without fire (shown by Figure 6a), the values float differently from the values (indicated by Figure 6b) in the period between 2:45 pm and 3:45 pm with fire.

Approach A1 can then perform smoothing of these signals by comparing the difference between the previously read values and the actual values. Thus, A1 will be able to identify any anomaly around the modules, which, consequently, could be a fire occurrence. This difference can be noticed in the graph of Figure 7, which shows the difference between the raw data and the exponential smoothing with t=1 and t=2.

The difference between these signals could be calibrated through adjustments in the weight variable ω. By adjusting ω, the difference between the signals allows each module firmware to compare the previously stored signals without uploading them from the server. Then, each module can trigger a message to the server with the sensor values only when an anomaly is detected. The difference can be obtained from Equation (Equation 2): (2)$$\begin{array}{cc} Y_{1} & =\omega X_{1}+\left(1-\omega \right)Y_{0},\\ Y_{2} & =\omega X_{2}+\left(1-\omega \right)Y_{1},\\ Dif_{A1} & =\left|Y_{1}-Y_{2}\right| \end{array}$$

This equation emphasized the importance of one data over another when its weight value is increased. Conversely, if the ω is reduced, the significance of one data over another will decrease. The weight variable can assume a lot of values between 0 and 1, and, therefore, the following section will point out possible values for ω based on fire tests and readings obtained from various modules.

### 4.2. Moving Average Based Algorithms

The Simple Moving Average (SMA) is a calculation for analyzing data points by creating a series of averages from different sets of data [49]. The idea behind moving averages is that observations which are nearby in time are also likely to be close in value. Then, taking an average of the points near an observation will provide a reasonable estimate of the trend cycle at that observation [50]. The calculation is achieved by taking the average of a specific number of data points (e.g., the last 10 data points) and then plotting that average as a single point. This moving average becomes a constantly updating average as new data points come in.

The SMA is used to smooth out short-term fluctuations and to highlight longer-term trends or cycles [51]. Given a set of *n* samples $X_{i}$, that represents the value collected by the sensor at the time *i*; the current moving average is expressed by Equation (Equation 3), where i=t,t−1,…,t−n+1 and *t* is a given moment.
(3)$$SMA_{t}=\frac{1}{n}\sum_{i=1}^{n}X_{i}$$

The term moving average is used to describe this procedure because each average is computed by dropping the oldest sample and including a new one [50]. In this sense, this work uses a simple moving average in the algorithm A2 to compare the difference between the actual data with stored averages’ values. In this case, a range of *n* samples is considered to evaluate a windows function that finds fire ignitions using a moving average. The following equation defines the error associated: (4)$$Dif_{A2}=\left|SMA_{t+n}-X_{1}\right|$$

Suppose an abrupt variation in the stored data (by each module) occurs. Such a situation is depicted in Figure 8. In this case, the moving average will be affected, getting out of the previous pattern. Hence, the difference between actual data with a simple moving average is considered to evaluate the presence of fire in the monitored region, as presented by the following condition. If $Dif_{A2}>D$ (with D>0), it could indicate the possibility of fire ignitions near a module. Thereupon, the data should be immediately transmitted to the cloud system. On the other hand, if the $Dif_{A2}\leq D$, the module will not send the message and consequently, it will help to reduce the volume of messages. To find the best values of *n* and *D*, which both need to respect the memory available in ATmega328, the following section is dedicated to testing values’ ranges to find them.

A variation of algorithm A2 is implemented in the third algorithm, denoted by A3. In the algorithm A3, the comparison is not made by looking for the actual data, as in algorithm A2. However, the comparison is made between two SMAs with different intervals (*n*), as described by Equation (Equation 5). In this sense, for each of the SMAs, two constants, α and *k*, are multiplied by the *n* from Equation (Equation 3), they are α and *k*, respectively. Thus, calculating $Dif_{A3}$ allows us to estimate the difference between two signals with the historical data collected by each module sensor (shown in Figure 9). Similar to previous approaches, it is expected that sensor identifies anomalies with the algorithm A3 and sends the data when a risk arises.
(5)$$Dif_{A3}=\left|SMA_{t+\alpha n}-SMA_{t+kn}\right|$$

When selecting different values for parameters α and *k*, consequently, different values for $Dif_{A3}$ will appear. Therefore, the next section will be dedicated to finding values for these three parameters according to the data acquisition performed with a set of modules during the days with and without fire tests.

### 4.3. Least Squares Based Algorithm

The Least Squares Strategy (LSS) is a common approach used to filter sensor signals [52,53]. It is a type of linear regression that uses the least squares method to determine the line of best fit for a data set. The strategy is used to remove any noise or errors that may be present in the signal. It works by minimizing the sum of the squares of the residuals (the difference between the data and the line of best fit) to obtain the best possible fit with the given data. The LSS can be used to filter out high-frequency noise and reduce the signal-to-noise ratio of the signal.

In the standard formulation of LSS, a set of *n* pairs of observations $\left\{x_{i},y_{i}\right\}$, is used to find a function relating the value of the dependent variable *y* to the values of an independent variable *x* [54]. Similar to previous algorithms A2 and A3, i=t,t−1,…,t−n+1 where *t* is the current moment. In the sensor problem, the number of samples is considered the independent variable *x*, and the value collected by each sensor is defined as the dependent variable *y*. The LSS defines the estimate of intercept α and the slope β of the regression line using the previous samples, which minimizes the sum of the squares between the sample measurements and the model [54]. The $\bar{x}$ and $\bar{y}$ are the mean of the $x_{i}$ and $y_{i}$ values used to predict the least squares function, respectively. Thence, the value prediction is given by Equation (Equation 6).
(6)$$\begin{array}{cc} \alpha & =\bar{y}-\beta \bar{x}\\ \beta & =\frac{\sum_{i=1}^{n}\left(x_{i}-\bar{x}\right)\left(y_{i}-\bar{y}\right)}{\sum_{i=1}^{n}{\left(x_{i}-\bar{x}\right)}^{2}}\\ y_{t+1} & =\alpha +\beta x_{t+1} \end{array}$$

The algorithm A4 proposed to optimize data transmission considering three previous samples to define a linear equation that minimizes the sum of the squares between the sample and the linear function value. In this way, the linear function represented by (Equation (Equation 6)) is used to predict the next sample (in this case, the 4th sample). In addition, this value is compared with the actual value ($x_{1}$) collected at the time t=4.
(7)$$Dif_{A4}=\left|y_{t+1}-x_{1}\right|$$

When the error between the value predicted $y_{t+1}$ and the actual value collected by the sensor, $x_{1}$, is bigger than *D*, as presented at expression (Equation (Equation 7)), the data should be immediately transmitted to the cloud, due to fire suspicion. Figure 10 represents the main idea graphically.

## 5. Results

A wildfire is created by accidentally burning vegetation in a wild or uncontrolled environment. Natural fire can cause panic and be challenging to contain; in addition, it could occur in remote areas. Natural fire also has some unpredictability in its initial phase. Therefore, controlled fire testing can be used to assess the readiness of emergency personnel and resources and to identify potential gaps in response plans or protocols. It assesses communication, organization, and coordination between incident command, emergency operations centers, fire departments, and other response agencies. Real fire simulations can also evaluate the effectiveness of existing wildfire management strategies and develop or refine future fire strategies and plans. The tests generally consist of a simulated forest fire scenario with predetermined objectives, a timeline, and a set of predefined parameters.

The tests used in this work are named Wildfire Simulation Events (WSE). It is a test designed to simulate an emergency forest ignition situation from the module’s point of view. The WSE’s objective is to test the behavior of the data through a bonfire in a field surrounded by trees (a scenario similar to the one found in Serra da Nogueira). In addition, the WSE is performed with the assistance of a firefighter in the fire’s containment in case the fire gets out of control.

Before carrying out the test, it is essential to ensure that all safety measures are taken. This includes choosing a selected area free of flammable materials (in this case, dry biomass) and providing all necessary firefighting equipment is available and working correctly. It is also vital to ensure that all observers are at a safe distance away from the fire. Combustion should be as close as possible to that found in situations of a natural forest fire; that is, during the execution of the WSE, only forest biomass should be used, such as branches, sticks, trunks, weeds, etc. Therefore, during the WSE, for security reasons, the following checklist was prepared:**Prepare the Area**: Remove all flammable waste, cut dead grass and brush, and ensure the fire pit is at least 4 meters away from any structures, combustibles, and vegetation;**Prepare the Fire**: Place a stone ring or other fire-resistant barrier around the fire. Make sure the ring is flat and securely in place;**Light the fire**: Gather local firewood and dry kindling. Use only a lighter or matches to light the fire. Write down the WSE start time (visually);**Monitor the fire**: Observe it from a distance of at least 5 meters and ensure it is contained and not spreading;**Put out the fire**: When the fire is no longer needed, use a shovel or bucket of water to put out the fire. As indicated by the firefighter, do not exceed 60 min. Note the time the WSE ends;**Cleanup**: Ensure all embers, ashes, and debris are removed from the area and disposed of properly.

After all the security measures were checked, and with the approval of the firefighter, ten sensor modules were fixed around the bonfire in the prepared area. The prepared area has four trees arranged in a circle, ideal for keeping all modules at a relative distance. The scenario created for WSE can be seen in Figure 11.

With the scenario arranged in a circular format, it is expected that the data acquisition by each module has similar behavioral characteristics. For example, the temperature sensitivity should be relatively the same between modules 2, 5, 7, and 9, as these four are under similar fire heights and distances. The distances of each module from the ground and the bonfire can be seen in Table 1. Overall, 4 WSEs were performed at different times of day and night.

All modules have the same firmware installed in their microcontrollers, except for the identification of each one. This way, it is possible to verify the data individually and guarantee in real time that all of them are sending data during the WSE execution. In addition, all ten modules remained fixed to the trunk of their respective trees for at least 20 days, working uninterruptedly. This makes it possible to analyze the differences between days with and without the occurrence of WSE. Due to the large amount of WSN data generated by ten modules every 60 s over 20 days, the following Table 2 shows an example of each module’s dataset.

With the WSN data from the ten modules stored on the server, it is possible to verify the approaches proposed in the previous section (A1, A2, A3, and A4). The following subsections will be dedicated to describing the results obtained from each of the approaches to identify behavioral anomalies after the data read moment.

### 5.1. A1 Results

Testing the A1 approach focuses on trial and error on different ω and *D* variables to see which one fits best in Equation (Equation 2). These variables can be changed in a given increment to verify the A1’s performances during the WSE intervals. Thus, when finding the values for these variables, the flame sensors’ signals could be filtered to avoid unnecessary transmissions (or less as possible). This can be done by manually adjusting the variables or using a computer program to automate the process.

The test plan to evaluate the performance of the A1 algorithm starts with the interval from 0 to 10 with an increment of 1 for the *D* values. In addition, the range from 0 to 1 with an increment of 0.1 is used for the ω values. Thus, Algorithm 1 summarizes the implementation of the code to perform this test.

After running the test, each increment in each of the variables will generate a separate column of data for $DROWSIER_{A1}$. Therefore, it is possible to compare each of these columns (True and False values) with the WSE column (True and False values). In this way, it is possible to identify how much each variable alerted the start of a fire when True & True happen simultaneously between these two columns of data. Considering only the WSE data, it is possible to perform a ranking of accuracy for each variable relating the amounts of alerts generated by quantities of data with fire, that is, making the proportion of alerts/fire. Figure 12a pointed out the result obtained through this relation.

When analyzing the result obtained through the first iteration, it is noted that the *D* values close to 10 have lower efficiency for any ω values. On the other hand, when *D* values get closer to 0, the efficiency of A1 increases. Therefore, a second iteration was carried out to analyze the behavior in this region of values for *D*. In this second iteration, the range from 0 to 1.5 was used in the variable *D* with an increment of 0.1, and the range of ω was maintained. The result obtained in this iteration can be seen in Figure 12b.
**Algorithm 1:** Pseudocode algorithm for testing the parameters of Exponential smoothing algorithm—A1.   i←1   **while**
i≤10
**do****Require:** Load data Node *i*     ω←0.1     D←1     **while** D<10 **do**         **while** ω<1.0 **do**            $Dif_{A1}=\left|Y_{1}-Y_{2}\right|$                        ▹ from Equation (Equation 2)            **if** $Dif_{A1}>D$ **then**                $DROWSIER_{A1}=True$            **else**                $DROWSIER_{A1}=False$            **end if**            ω←ω+0.1         **end while**         D←D+1     **end while**     i←i+1**end while**

### 5.2. A2 Results

The A2 approach uses short historical data to predict temperature ranges for a given sampling. Therefore, it is necessary to identify what this amount of sample will be considered. This can be done using a computer program to iteratively adjust the *n* and *D* values until Equation (Equation 4) is solved. With this, it is expected to find values for *n* and *D* that can distinguish critical temperature variations before the data is sent by each module. In this sense, for the A2 algorithm, a test plan was made similar to the previous one. However, in this test is used the same range from 0 to 10 with an increment of 1 in *n* and *D*, as demonstrated in Algorithm 2.
**Algorithm 2:** Pseudocode algorithm for testing the parameters of SMA algorithm—A2.  i←1  **while**
i≤10
**do****Require:** Load data Node *i*     t←0     n←1     D←1     **while** D<10 **do**         **while** n<10 **do**            $Dif_{A2}=\left|SMA_{t+n}-X_{1}\right|$                        ▹ from Equation (Equation 4)            **if** $Dif_{A2}\geq D$ **then**                $DROWSIER_{A2}=True$            **else**                $DROWSIER_{A2}=False$            **end if**            n←n+1         **end while**         D←D+1     **end while**     i←i+1**end while**

A separate column of $DROWSIER_{A2}$ data for each increment in each of the variables was generated after running the test. Then, as the same way is done before, it is possible to compare each of these columns with the WSE column. In this sense, Figure 13 demonstrates the results obtained through relation alerts/fire from the A2’s output.

When using the range from 0 to 10 with an increment of 1 for *n* and *D*, it is possible to notice that the effectiveness of A2 is low when *n* and *D* have values close to 10. In addition, when *D* has values around 10 and *n* has values proximate to 0. On the other hand, when *D* has values nearest to 0 and *n* has values close to 10, the efficiency of A2 increases. However, there is a gap between the values of 0.2 and 0.6 on the Z axis (Figure 13a).

A new test was run to mitigate the gap from 0.2 to 0.6 in the Z-axis, or at least to find better approximations. Therefore, a second iteration was performed to analyze the behavior of A2 with more specific values between zones 0 to 1 of *D* and maintain the same range for *n*. The second iteration ranges from 0 to 0.9, and an increment of 0.05 for *D* is shown in Figure 13b.

### 5.3. A3 Results

As mentioned in the previous section, the A3 approach uses the difference between two SMAs to identify abrupt changes in flame sensor data before sending it to the server. Therefore, this approach focuses on distinguishing the behaviors in flame sensor data through signal filtering. The identification is made through the amplitude of the difference between two SMAs, so the A3 must be able to compare this amplitude with data with and without WSE. To make the A3 able to distinguish between forest and non-forest ignition moments, it is necessary to find the values of αn and kn that satisfy Equation (Equation 5).

The test plan for finding αn and kn values is summarized by Algorithm 3, where different ranges for these values are defined. Then, through the implementation of the algorithm, computational tests are performed with the range from 0 to 10 and increment of 1 for both αn and kn values. When running these values for each variable, different values are expected for $Dif_{A3}$. For this reason, a *D* range with values from 0 to 10 with an increment of 1 was also applied. Outliers were considered when the values chosen for αn and kn were the same since this calculation always generates null values.

After executing this test, similarly to what was done in the previous tests, the accuracy rate can be found by comparing each of the A3’s outputs with the values contained in the WSE column. The accuracy rate plot based on the values of *D*, αn, and kn can be seen in Figure 14a.

Some values obtained through the first iteration can be treated as outliers. When the accuracy rate is very close to or equal to zero, it could mean that the used values in αn and kn parameters could not identify any variations in the data (difference between raw and smoothed) with the occurrence of WSE. Therefore, the graph in Figure 14b demonstrates the behavior of the A3 approach for accuracy values greater than 0.5. A second iteration is unnecessary to perform since the ranges of values for *D*, αn, and kn used return high accuracy rates.
**Algorithm 3:** Pseudocode algorithm for testing the parameters of Differences between two SMAs algorithm—A3.  i←1  **while**
i≤10
**do****Require:** Load data Node *i*     t←0     αn←1     D←1     kn←1     **while** αn<10 **do**         **while** D<10 **do**            **if** αn≠kn **then**                **while** kn<10 **do**                    $Dif_{A3}=\left|SMA_{t+\alpha n}-SMA_{t+kn}\right|$                      ▹ from Equation (Equation 5)                    **if** $Dif_{A3}\geq D$ **then**                        $DROWSIER_{A3}=True$                    **else**                       $DROWSIER_{A3}=False$                    **end if**                    kn←kn+1                **end while**            **end if**            D←D+1         **end while**         αn←αn+1     **end while**     i←i+1 **end while**

### 5.4. A4 Results

The identification of variations in the flame sensors data when a forest ignition arises could also be verified with the A4 approach. As previously described, Equation (Equation 7) can be used for this identification. However, it still remains to be seen which values to use for $Dif_{A4}$. Then, through the implementation of Algorithm 4, it is possible to apply a range of values to obtain the results of A4.

A range between 0 and 10 incremented with 1 is used as the test plan to identify the value of *D*. With the insertion of these values in computational processes, it was possible to generate the graph in Figure 15a. This graph demonstrates the accuracy of the A4 in identifying abrupt changes during data acquisition of the flame sensors with WSE.

Based on the A4’s output presented in Figure 15a, it is noted that the more the *D* value increases, the less accuracy is obtained for forest ignition alerts. Therefore, a second iteration was performed with a range from 0 to 0.9 and an increment of 0.1. Figure 15b demonstrates the result of the A4’s accuracy with this last range of values.
**Algorithm 4:** Pseudocode algorithm for testing the parameters of Least Squares algorithm—A4.  i←1  **while**
i≤10
**do****Require:** Load data Node *i*     t=3     D←1     **while** D<10 **do**         $Dif_{A4}=\left|y_{t+1}-x_{1}\right|$                        ▹ from Equation (Equation 7)         **if** $Dif_{A4}\geq D$ **then**            $DROWSIER_{A4}=True$         **else**            $DROWSIER_{A4}=False$         **end if**         D←D+1     **end while**     i←i+1 **end while**

### 5.5. Final Considerations

After all the runs in the previous subsections, it is necessary to choose the best parameters for each approach. In this sense, the combination of all approaches setting up with the best values will result in a final algorithm DROWSIER. Therefore, in this subsection, all experiments carried out with the DROWSIER algorithm are described.

The combination of the four algorithms, the DROWSIER filter, can be defined using the five best parameters selected. As previously described, the score criterion among the parameters was due to each algorithm alerting when the WSE was performed (the beginning of a forest fire). Therefore, other possible alerts (false alerts) outside this range are not yet considered. Thus, the five best results of each approach were selected to verify the DROWSIER filter. Table 3 shows each of the chosen values and their respective accuracies.

Each approach can be configured with its best five parameters through computational tests. Thus, five combinations are proposed to test the accuracy of each of them working together. The first combination (C1) is focused on generating alerts during WSE when all four approaches (A1+A2+A3+A4) identify an anomaly. Similarly, the other four combinations are set to remove each approach one at a time. That is, the second combination (C2) uses the union of A2, A3, and A4; A1, A3, and A4 generate the third combination (C3); the penultimate combination (C4) is carried out through the combination of A1, A2, and A4; the last combination (C5) removes approach A4 from the analysis, checking A1, A2, and A3.

Considering the equations demonstrated in Section 4, it is expected that some approaches can adapt depending on the WSE. For example, in the case of approaches that use averages to identify anomalies during data acquisition. Therefore, the five combinations were run with different WSE time intervals. Thus, in addition to the entire WSE time interval (named All time), the intervals of 5, 10, 15, and 20 min (called $5^{\prime }$, $10^{\prime }$, $15^{\prime }$, and $20^{\prime }$, respectively) were chosen. These last four intervals were configured according to the start of each WSE. Thus, the $5^{\prime }$ had only the initial 5 min of each of the four WSEs performed. The same was true for $10^{\prime }$, $15^{\prime }$, and $20^{\prime }$. Figure 16 demonstrates all the performances of each combination (C1−C5) during the mentioned intervals.

The C1, considering all WSE intervals used, obtained data anomaly results with a maximum accuracy value of 0.60 and a minimum value of 0.45. On the other hand, also considering all WSE intervals, C2 achieved accuracy results between 0.69 (maximum) and 0.57 (minimum). The third combination (C3) reached a minimum accuracy of 0.61 and a maximum of 0.69, with its best parameters for all WSE intervals mentioned above. After performing C4, the maximum and minimum accuracy values during all WSE intervals are 0.60 and 0.45, respectively. In addition, the C5, when analyzed within all WSE intervals, has an accuracy of 0.73 at its maximum and 0.54 at its minimum. All the maximum and minimum accuracies of each combination can be seen in Table 4, as well as the parameters used in each approach and the distinct WSE intervals.

Two combinations can then be responsible for composing the DROWSIER filter, and they are C3 and C5. In the case of C3, there is a difference of 0.08 between its maximum and minimum accuracy. On the other hand, under the same comparison of maximums and minimums, there is a difference of 0.19 in the C5 combination. However, the last combination has a higher accuracy, with an advantage of 0.04 over the maximum C3’s accuracy.

A possible tiebreaker between C3 and C5 could be performed by analyzing the false alerts of each one. However, comparing the false alerts between them would be unfair since they have different accuracy values. Thus, it is possible to score C3 from C5 (or vice versa), individually taking the ranges between the maximum and minimum values of its false alerts. However, both have equal extreme values of false alerts, 0.42 for C3 and 0.50 for C5.

As mentioned before, the ATmega328 microcontroller does not have large amounts of memory to store. Therefore, keeping previously read values could be a criterion for choosing between C3 and C5. Among the four approaches used in this work, the one that consumes the most memory is A3. Since this approach needs to store *n* samples of the five flame sensors in the modules. The column named Variables, in Table 4, presents that both C3 and C5 need to be configured with kn=6 (one of the three parameters of A3). Therefore, these two combinations have the same amount of space stored in the microcontroller employed.

Regarding WSE time intervals, a strategy could be used to choose the best combination between A1, A2, A3, and A4. The combinations C3 and C5 could be selected based on the main objective of this work, which is to find a digital filter that can reduce the energy consumption resulting from unnecessary sending. Therefore, the combination that grants the nodes with the longest time interval could be chosen. This is because the longer the time interval that each module sends data, the less each one of them will consume from its own battery. In this sense, the combination C5 has an advantage over C3.

With the choice of the $15^{\prime }$ WSE interval, the base firmware of each module could be configured to deep sleep between intervals of 60 s (same sampling interval of the data used throughout this work). Then, between each 60-second break, the DROWSIER filter output could be checked using the C5 combination. In case there is an anomaly identified by DROWSIER during data acquisition similar to that performed during WSE, the actual data from each sensor will be immediately sent to the server. Otherwise, the module would go back to its deep sleep. Accordingly, this cycle could be repeated until reaching the number 15 (completing 15 min). At the end of 15 min, each module must send its current data.

After implementing the DROWSIER in the sensor module’s firmware, it is possible to notice the difference in consumption during 60 s. Figure 17 displays the graph showing the current reading during the data acquisition cycle of the node model. Therefore, comparing the consumption shown in Figure 4 and Figure 17, there was a reduction in the average current read from 672.30 μA to 356.86 μA. By using the C5 configuration, each module can reduce almost 53% of its consumption when is no request to send data.

For the server to distinguish whether a message is coming from the DROWSIER or not, a variable must be chosen to be sent along with the sensors’ data. Future analyses could benefit from this information. Note that this variable cannot decide whether there is a forest ignition. As previously mentioned, machine learning algorithms will work in the cloud to interpret the data received by each node and, consequently, generate alerts to be sent to the authorities.

## 6. Conclusions

The SAFe project aims to acquire data from the environment and transfer it to central operations. The data includes information such as temperature, humidity, and flame sensor. The data will be analyzed to provide an early warning system for fire alerts. The system will also monitor the forest conditions to detect environmental changes that may indicate a high risk of fire ignition. The SAFe project will contribute to the improvement of fire alert operations, providing timely information about forest conditions and the risk of fire ignition. This will enable firefighters and civil protection to act quickly and efficiently in emergencies.

Regarding communication, the SAFe project uses a wireless communication system based on the LoRaWAN protocol that enables real-time data transmission. In this sense, by spreading several sensor modules (creating a WSN) in a forest, each node can detect a fire ignition as soon as possible. However, the limited resources of the LoRaWAN, in terms of bandwidth to attend to a high number of nodes, require smart techniques in data transmission. Additionally, since each module is placed in remote zones, the lifetime of batteries needs to be improved to avoid early maintenance.

Based on that, this work proposed applying four algorithms to optimize the communication between the sensor modules and the cloud system: the exponential smoothing algorithm (A1), two strategies denoted as the moving average-based algorithms (A2 and A3), and the least squares based algorithm (A4). The best parameters’ values (of each approach) were recognized by testing each algorithm individually using stored sensor data with a four WSE. After that, five combinations between these four algorithms were tested with several WSE intervals.

A filter named DROWSIER was implemented with the best combination of these four algorithms. The C5 was chosen after comparing it with other combinations. Its high accuracy rate of 0.73 and a false alert rate of 0.5 in a WSE interval of $15^{\prime }$ indicates that it can reduce almost 53% of battery consumption during data transmission (when is no demand to send data). Consequently, the best-identified time interval leads to a decrease in LoRaWAN network usage.

In terms of future work, it is considered to check the usage of the LoRaWAN network and the battery after the implementation of DROWSIER in real cases. Using the same WSE, or more closely as possible, future tests can define the performance of a WSN using the DROWSIER filter. Moreover, comparing the battery’s discharge over a long time in modules with and without DROWSIER. New variations in each algorithm’s equations can also be checked. For example, the A1 could be tested using double or triple exponential smoothing. Other methods could be tested, such as Savitzky–Golay filters for smoothing signals using low-degree polynomials and compared with raw data. Moreover, performing the WSE in other scenarios could represent the possibility of applying the SAFe project in other regions. Then, the subsequent studies can elaborate on other validations for this purpose. Moreover, the sample rate of each sensor can be variate; for example, time intervals lower than 60 s could be found using the data acquisition for one by one sensor or not (regarding the usage of the LoRaWAN Gateway), thus, some future work can prove it.

## Figures and Tables

**Figure 1 sensors-23-01282-f001:**
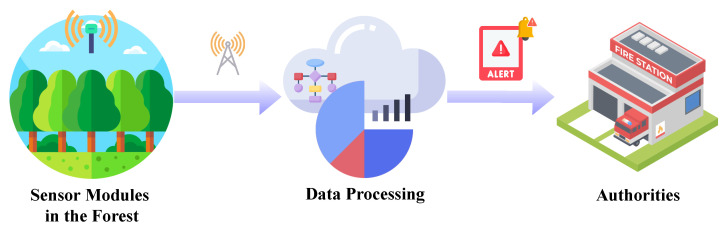
High-level architectural view of SAFe project.

**Figure 2 sensors-23-01282-f002:**
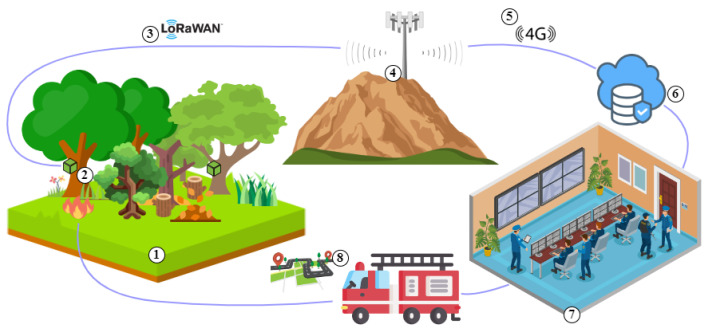
SAFe architecture: ➀ Monitored region, ➁ Wireless sensor module, ➂ Data transmission using LoRaWAN, ➃ LoRaWAN gateway, ➄ Mobile internet, ➅ Cloud server, ➆ Control center, and ➇ Fire ignition early attack.

**Figure 3 sensors-23-01282-f003:**
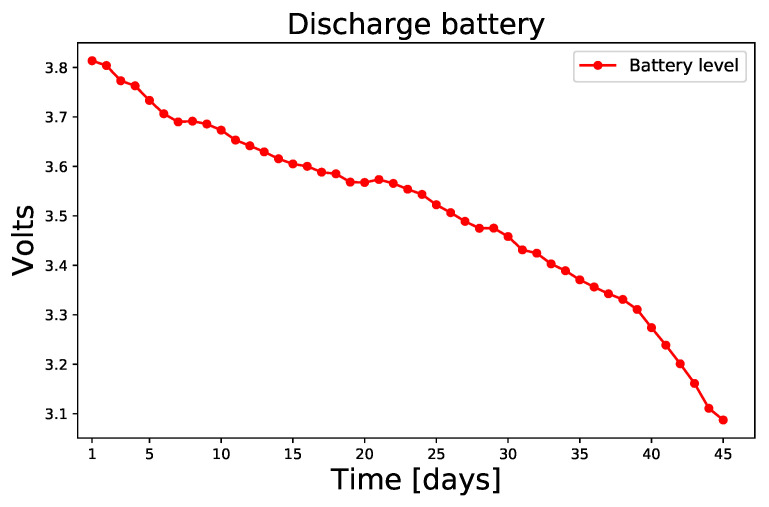
Discharging the SAFe module battery until the module cannot send data.

**Figure 4 sensors-23-01282-f004:**
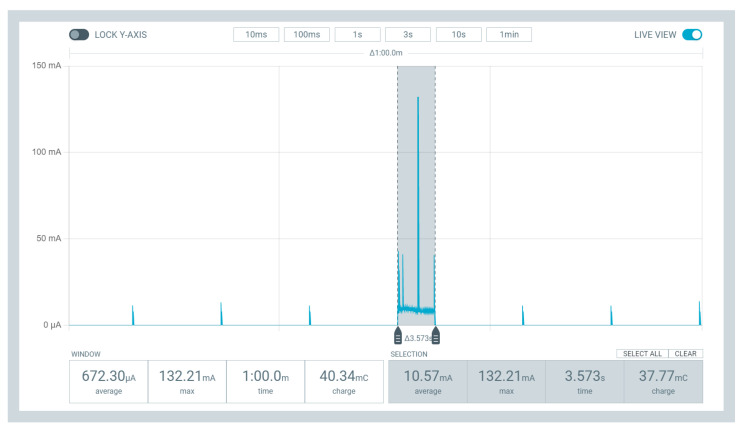
Example of sensor modules’ consumption.

**Figure 5 sensors-23-01282-f005:**
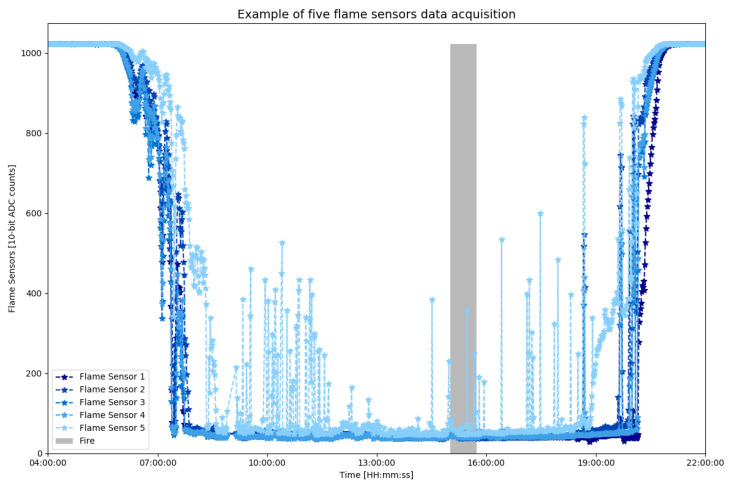
Flame sensors raw data from an example of a sensor module attached to a tree trunk during a fire assay.

**Figure 6 sensors-23-01282-f006:**
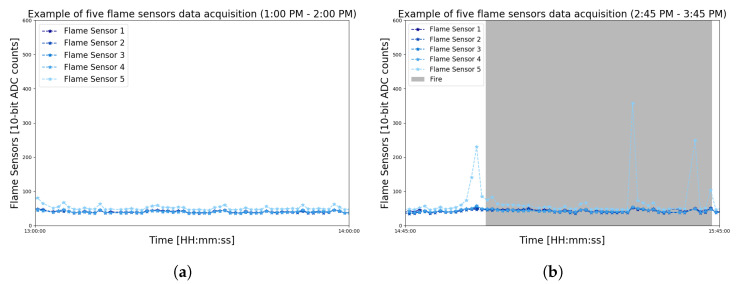
Zoomed view of the raw flame sensor data to compare the noises after the data acquisition of a fire test. (**a**) Noise in flame sensors without fire. (**b**) Noise in flame sensors with fire.

**Figure 7 sensors-23-01282-f007:**
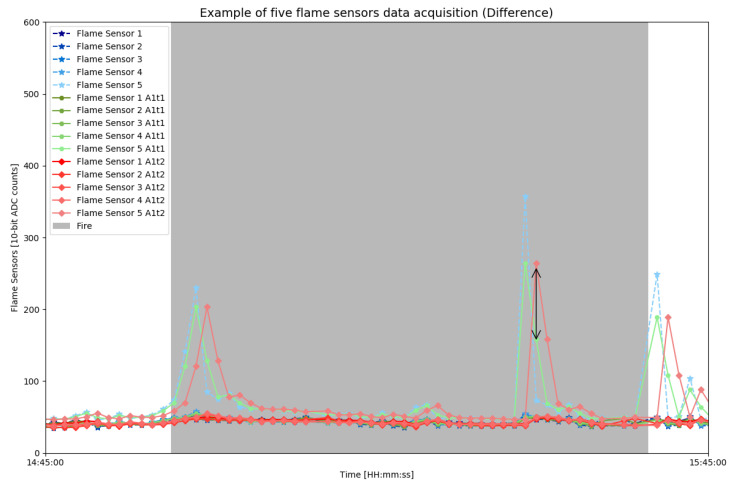
Difference plotting produced by the A1 approach configured for t=1 and t=2. It is possible to graphically visualize the difference between raw and smoothed data at two time intervals.

**Figure 8 sensors-23-01282-f008:**
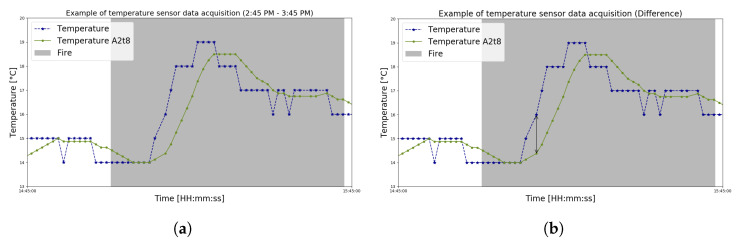
Zoomed plotting of the sensor module’s temperature data that exemplifies the difference proposed by approach A2 through n=8. (**a**) The raw and smoothed temperature sensor during a fire test. (**b**) Black arrow with unknown *D* size marking the difference between raw and smoothed data.

**Figure 9 sensors-23-01282-f009:**
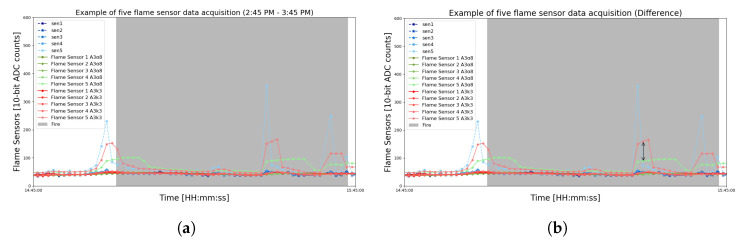
Zoomed plotting of the sensor module’s flames data that illustrates the difference proposed by approach A3 through αn=8 and kn=3. (**a**) The raw and smoothed flame sensor during a fire test. (**b**) Black arrow with unknown *D* size marking the difference between raw and both smoothed data.

**Figure 10 sensors-23-01282-f010:**
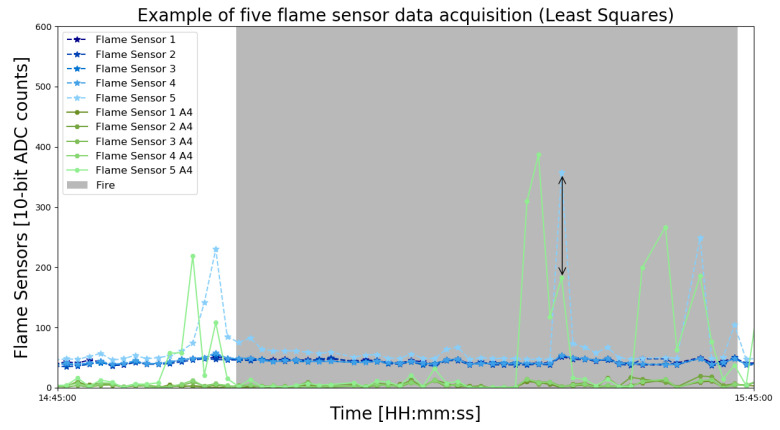
Difference plotting proposed by the A4 approach configured for t=3. It is possible to visualize the difference between raw and predicted data graphically.

**Figure 11 sensors-23-01282-f011:**
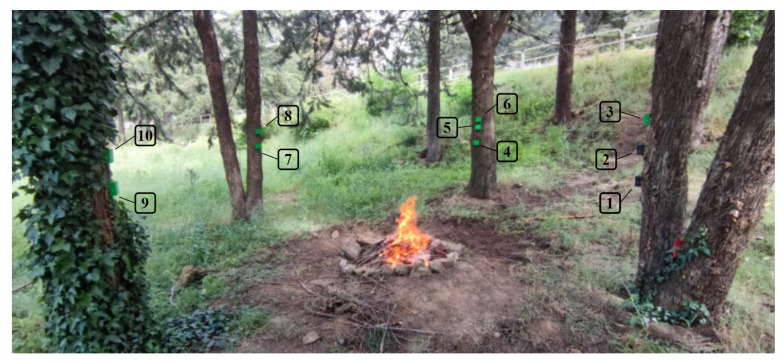
Scenario created to elaborate the WSE using ten nodes to form a SAFe’s WSN. Test carried out to identify the behavior of the WSN’s data acquisition during real-fire situations. All safety measures were followed, and a firefighter supervised all WSEs.

**Figure 12 sensors-23-01282-f012:**
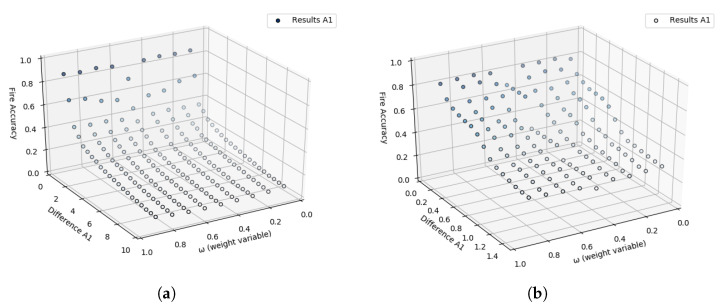
Graph visually demonstrating the result of the A1 approach with the mentioned intervals for *D* and ω from 0 to 1.0 stepping 0.1. (**a**) A1’s results with *D* range 0 to 10 stepping 1. (**b**) A1’s results with *D* range 0 to 1.5 stepping 0.1.

**Figure 13 sensors-23-01282-f013:**
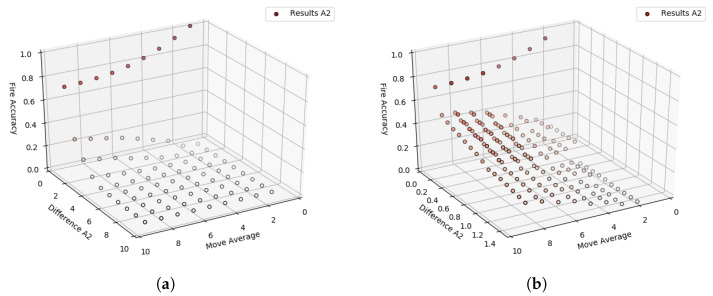
Graph visually demonstrating the result of the A2 approach with the mentioned intervals for *D* and *n* from 0 to 10 stepping 1. (**a**) A2’s results with *D* range 0 to 10 stepping 1. (**b**) A2’s results with *D* range 0 to 0.9 stepping 0.05.

**Figure 14 sensors-23-01282-f014:**
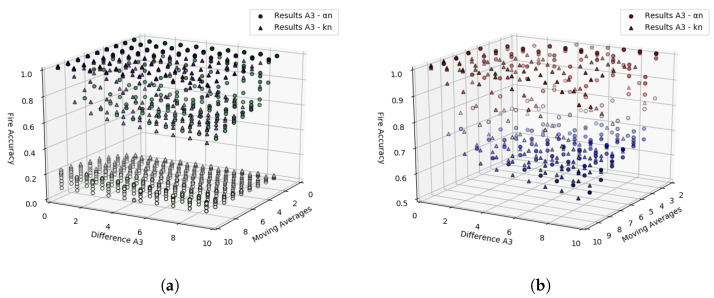
Graph visually demonstrating the result of the A3 approach with the range from 0 to 10 stepping 1 to both αn and kn. (**a**) A3’s results with αn and kn range 0 to 10 stepping 1. (**b**) A3’s results after dropping the outliers.

**Figure 15 sensors-23-01282-f015:**
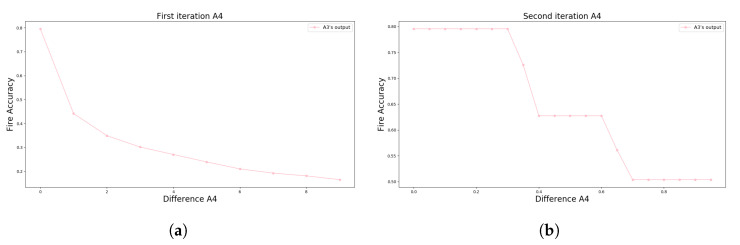
Graph visually demonstrating the result of the A4 approach with the mentioned range for *D* using t=3. (**a**) A4’s results with *D* range from 0 to 10 stepping 1. (**b**) A4’s results with *D* range from 0 to 0.9 stepping 0.1.

**Figure 16 sensors-23-01282-f016:**
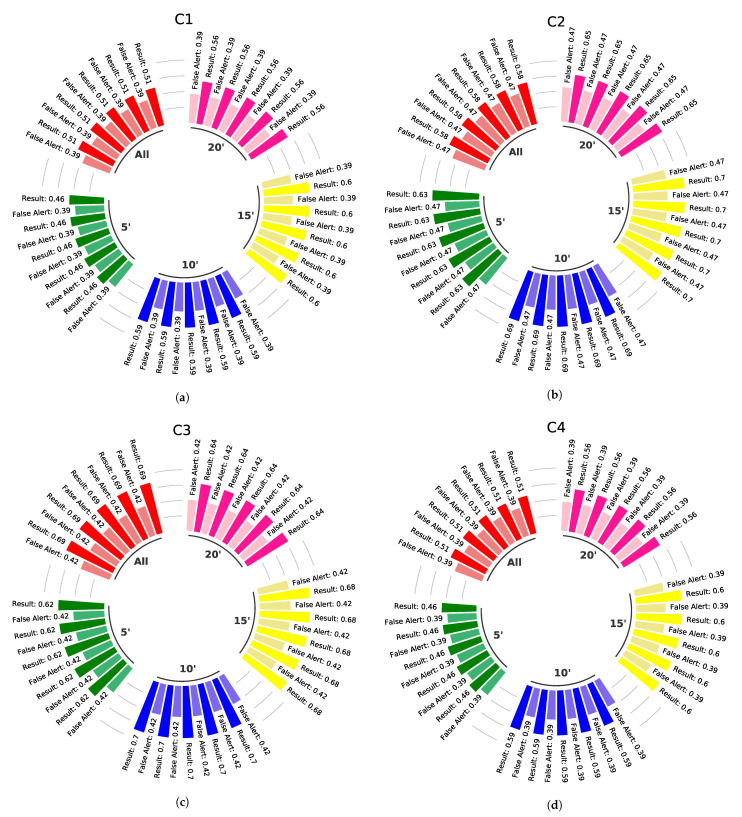
Performance of combinations at different time intervals between each of the proposed approaches when it’s configured with their best variable values. (**a**) Performance of C1. (**b**) Performance of C2. (**c**) Performance of C3. (**d**) Performance of C4. (**e**) Performance of C5.

**Figure 17 sensors-23-01282-f017:**
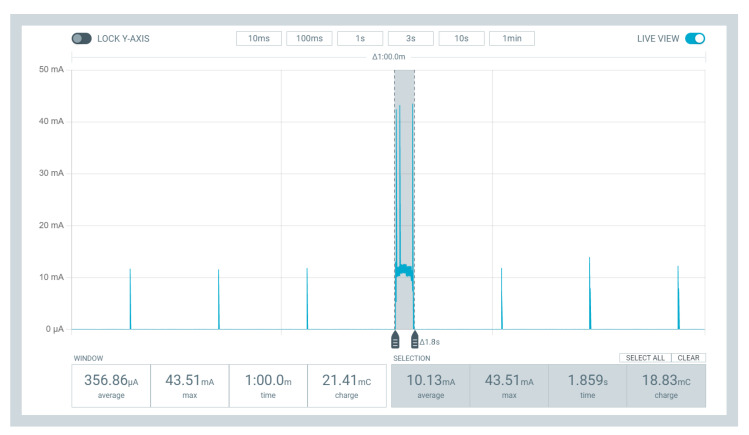
Example of sensor modules’ consumption with DROWSIER filter.

**Table 1 sensors-23-01282-t001:** Distances from the ground and from the fire of each of the sensor modules.

Module	Ground [m]	Bonfire [m]
Node 1	1.12	3.70
Node 2	1.37	3.70
Node 3	1.60	3.70
Node 4	1.12	3.60
Node 5	1.37	3.60
Node 6	1.60	3.60
Node 7	1.37	3.50
Node 8	1.60	3.50
Node 9	1.37	3.90
Node 10	1.60	3.90

**Table 2 sensors-23-01282-t002:** Demonstration of the dataset’s structure generated by a given sensor module stored on the server after the four WSEs.

Date [HH:mm]	Battery [V]	Humidty [%]	Temperature [ºC]	Flame Sensor 1	Flamse Sensor 2	Flame Sensor 3	Flame Sensor 4	Flame Sensor 5	WSE (Fire)
Day 1 00:01	3.82	75	10	42	45	42	55	40	False
Day 1 00:02	3.82	75	10	44	41	44	56	42	False
*…*	*…*	*…*	*…*	*…*	*…*	*…*	*…*	*…*	*…*
Day 5 12:02	3.63	56	19	46	45	46	64	46	True
Day 5 12:03	3.63	51	19	40	38	39	47	39	True
*…*	*…*	*…*	*…*	*…*	*…*	*…*	*…*	..	*…*
Day 20 23:58	3.48	70	17	1023	1023	1023	1023	1023	False
Day 20 23:59	3.48	71	17	1023	1023	1023	1022	1023	False

**Table 3 sensors-23-01282-t003:** Best five accuracies obtained and their respective parameters’ values of each approach previously described.

Approach	Variables	Values	Results with WSE
A1	ω;*D*	(0.7;0.1)	0.834630350
		(0.2;0.1)	0.834630350
		(0.3;0.1)	0.832684825
		(0.4;0.1)	0.831712062
		(0.8;0.1)	0.829766537
A2	*n*,*D*	(6;0.0)	0.727626459
		(6;0.0)	0.709143969
		(8;0.0)	0.696498054
		(8;0.05)	0.458171206
		(8;0.1)	0.458171206
A3	αn,kn,*D*	(5;6;0.0)	0.999027237
		(4;6;1.0)	0.999027237
		(3;6;3.0)	0.999027237
		(2;6;7.0)	0.999027237
		(2;4;6.0)	0.998054475
A4	*D*	0.2	0.795719844
		0.0	0.795719844
		0.1	0.795719844
		0.3	0.795719844
		0.6	0.627431907

**Table 4 sensors-23-01282-t004:** Maximums and Minimums of the performances of the five combinations with the variables identified for each case.

Combination	WSE Intervals		Variable		Result		False Alert ^1^	
	$Result_{Max}$	$Result_{Min}$	$Result_{Max}$	$Result_{Min}$	Max.	Min.	Max.	Min.
C1	$15^{\prime }$	$5^{\prime }$	A1 (ω=0.3; D=0.1) A2 (n=6; D=0.0) A3 (αn=4; kn=6; D=1.0) A4 (D=0.0)	A1 (ω=0.7; D=0.1) A2 (n=6; D=0.0) A3 (αn=5; kn=6; D=0.0) A4 (D=0.3)	0.60	0.45	0.60	0.38
C2	$15^{\prime }$	${}^{\prime }All^{\prime }$	A2 (n=6; D=0.0) A3 (αn=4; kn=6; D=1.0) A4 (D=0.0)	A2 (n=6; D=0.0) A3 (αn=3; kn=6; D=3.0) A4 (D=0.3)	0.69	0.57	0.46	0.47
C3	$10^{\prime }$	$5^{\prime }$	A1 (ω=0.7; D=0.1) A3 (αn=4; kn=6; D=1.0) A4 (D=0.2)	A1 (ω=0.7; D=0.1) A3 (αn=5; kn=6; D=0.0) A4 (D=0.0)	0.69	0.61	0.42	0.42
C4	$15^{\prime }$	$5^{\prime }$	A1 (ω=0.3; D=0.1) A2 (n=6; D=0.0) A4 (D=0.0)	A1 (ω=0.7; D=0.1) A2 (n=6; D=0.0) A4 (D=0.0)	0.60	0.45	0.38	0.38
C5	$15^{\prime }$	$5^{\prime }$	A1 (ω=0.3; D=0.1) A2 (n=6; D=0.0) A3 (αn=5; kn=6; D=0.0)	A1 (ω=0.2; D=0.1) A2 (n=6; D=0.0) A3 (αn=5; kn=6; D=0.0)	0.73	0.54	0.50	0.50

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
