# Peer review of "Data Acquisition Filtering Focused on Optimizing Transmission in a LoRaWAN Network Applied to the WSN Forest Monitoring System"

_sensors, 2023, doi:10.3390/s23031282_

Round 1

Reviewer 1 Report

1. Battery consumption reduction by 53% during data transmission is not justified in the analysis and results. 

2. Also, 50% of the false alert cases are too large, considering the severity of the application. 

3. Are the combination of algorithms tested or simulated for different forest fires like mild, medium, and severe cases?

4. What about the control signals exchanged between the Nodes and the LoRAWAN gateway? They also contribute to battery drain. Can they also be optimized? Please provide clarity regarding this.

5. Can a similar analysis be made by varying data transmission frequency from node to gateway? Any reason for making it 60 seconds? Is it fixed throughout the analysis, or can it be varied?

6. How will the control center know about the battery status? Any back engineering is done to deduce that?

7. How will the control center know about dead nodes with wholly drained batteries?

8. Is there a chance to charge the node's batteries by any mechanism?

9. Again, implementing the combination of algorithms on the node side, this again will consume more power. How is that being addressed?

Author Response

Dear Reviewer,

Thank you for your comments and suggestions. In the attachment is the Author's Reply.

Best regards.

Reviewer 2 Report

The authors employed a Wireless Sensor Network (WSN) for forest data acquisition to identify abrupt anomalies when a fire ignition starts. The authors used a low-power LoRaWAN network, and four algorithms were tested and calibrated along with real and monitored events from a wildfire. On validations, the developed filter achieves an accuracy rate of 0.73 with 0.5 possible false alerts. The proposed modules can save almost 53% of their batteries.

The implementation of LoRaWAN technology is an interesting topic; however, the authors must clarify and address the following points:

  1. In an abstract, the authors mentioned the employment of WSN and LoRaWAN technology and the testing of four algorithms. However, the contributions, like the employment of WSN, LoRaWAN technology and the testing of algorithms, are not significant enough to publish an article with eight to nine authors. The authors should mention a sentence or two about their novel contributions in the abstract.
  2. The framework proposed in this research work requires high scalability, therefore, authors are required to mention how the scalability factors will be handled in this research work.
  3. There is an inconsistency in using the words SAFe and SAFE.
  4. Figure 9 is not explained or mentioned anywhere in the text.
  5. What are the limitations and assumptions of the proposed research work?
  6. The discussion of results needs to be more comprehensive before the conclusion sections.
  7. The authors should be required to describe the complexity of the algorithms presented in the article, like in Algorithm 3, where three nested while loops are being used; if there is high complexity, then what will be the tradeoffs in this regard and how the authors will address such tradeoffs?
  8. The related work should include some more research on LoRaWAN technology, and a comparative analysis at the end of the related work is also required to be written.
  9. There is repetition of sentences, such as in a paragraph before Section 4.1, which has some sentences that are repeated in the article at other places like the abstract, introduction, and conclusion.
  10. The authors claimed to achieve accuracy and reduce power consumption but did not provide a reason in the results sections. 

Author Response

(The authors gave the same response as above.)

Reviewer 3 Report

Please see the attached comments.

Author Response

(The authors gave the same response as above.)
